# Overall Survival for Esophageal Squamous Cell Carcinoma with Multiple Primary Cancers after Curative Esophagectomy—A Retrospective Single-Institution Study

**DOI:** 10.3390/cancers14215263

**Published:** 2022-10-26

**Authors:** Ping-Chung Tsai, Ying-Che Ting, Po-Kuei Hsu, Jung-Jyh Hung, Chien-Sheng Huang, Wen-Hu Hsu, Han-Shui Hsu

**Affiliations:** 1Division of Thoracic Surgery, Department of Surgery, Kaohsiung Veterans General Hospital, Kaohsiung 813414, Taiwan; 2Division of Thoracic Surgery, Department of Surgery, Taipei Veterans General Hospital, Taipei 112201, Taiwan; 3School of Medicine, National Yang Ming Chiao Tung University, Taipei 112304, Taiwan

**Keywords:** esophageal squamous cell carcinoma, multiple primary cancers, head and neck cancer, overall survival

## Abstract

**Simple Summary:**

With a strong association between esophageal cancer and multiple primary cancers (MPCs), physicians encounter difficult decision making for appropriate treatment. By analyzing long-term survival for esophageal squamous cell carcinoma (SCC) associated with MPCs, patients with antecedent/synchronous cancers showed similar survival benefit in comparison with esophageal SCC only. The most common site of other primary cancers is the head and neck, which also showed no inferiority in overall survival (OS) among sites of MPCs. Once curative esophagectomy can be achieved, surgery should be selected as a potential therapeutic approach if indicated.

**Abstract:**

Background: Advances in surgical techniques and treatment modalities have improved the outcomes of esophageal cancer, yet difficult decision making for physicians while encountering multiple primary cancers (MPCs) continues to exist. The aim of this study was to evaluate long-term survival for esophageal squamous cell carcinoma (SCC) associated with MPCs. Methods: Data from 544 patients with esophageal SCC who underwent surgery between 2005 and 2017 were reviewed to identify the presence of simultaneous or metachronous primary cancers. The prognostic factors for overall survival (OS) were analyzed. Results: Three hundred and ninety-seven patients after curative esophagectomy were included, with a median observation time of 44.2 months (range 2.6–178.6 months). Out of 52 patients (13.1%) with antecedent/synchronous cancers and 296 patients without MPCs (control group), 49 patients (12.3%) developed subsequent cancers after surgery. The most common site of other primary cancers was the head and neck (69/101; 68.3%), which showed no inferiority in OS. Sex and advanced clinical stage (III/IV) were independent risk factors (*p* = 0.031 and *p* < 0.001, respectively). Conclusion: Once curative esophagectomy can be achieved, surgery should be selected as a potential therapeutic approach if indicated, even with antecedent/synchronous MPCs. Subsequent primary cancers were often observed in esophageal SCC, and optimal surveillance planning was recommended.

## 1. Introduction

Esophageal cancer is ranked sixth in mortality overall, being one of the most aggressive malignancies, which has differences in incidence and mortality rates between the sexes [1]. Despite the development of multimodal treatment, long-term survival rarely exceeds 50% after esophagectomy [2]. The majority of histological types of esophageal cancer are squamous cell carcinoma (SCC) in Asia, with the highest incidence rates from Eastern to Central Asia [3]. Clinical practice guidelines recommended that the presence of multiple primary cancers (MPCs) of the aerodigestive tract should be investigated, especially chronic exposure to common carcinogens such as tobacco and alcohol consumption [4]. Meanwhile, frequent development of simultaneous or metachronous cancer happens owing to carcinogenic effects. The incidence of MPCs associated with esophageal malignancy has been reported to be 14.5% to 35.9% [5,6,7]. The association of high risk of developing MPCs has been explained by the concept of “field cancerization”, which concerns repeated exposure of the epithelium of the upper aerodigestive tract to carcinogens [8,9].

As for independent malignant lesions detected before or at esophagectomy among MPCs, the most common site is the head and neck. An exceptionally strong association between esophageal cancer and head and neck cancer has been reported [10]. Patients with antecedent or synchronous malignancies with esophageal cancer have a significantly increased risk of developing subsequent malignancies, which has been considered an unfavorable outcome previously [10]. However, the developing screening strategy in consideration of encountering MPCs has been gradually improving, which might lead to the reduction of serious adverse events and the risk of death [11]. Long-term survival of patients with esophageal SCC associated with MPCs remains controversial, which might bring about a difficult therapeutic decision while encountering simultaneous MPCs. Therefore, we investigated long-term survival and prognostic factors of MPCs in patients with esophageal SCC treated by curative esophagectomy in our single-center institution.

## 2. Materials and Methods

### 2.1. Study Design and Eligibility Criteria

We retrospectively reviewed patients with esophageal SCC with or without second primary cancers at Veterans General Hospital (Taipei, Taiwan) between January 2005 and December 2017 and identified 544 patients. We excluded: (1) patients lost to follow-up in 2 years or missing data during surveillance; (2) patients with an initial M1 stage, R1/R2 resection, or intended salvage operation; (3) patients with surgery-related mortality; (4) patients with skip esophageal lesions; and (5) patients with more than a 5-year interval between esophageal SCC and metachronous cancer.

Preoperative staging work-up included a systemic physical examination, laboratory screening, esophagogastroduodenoscopy, bronchoscopy for tumors in upper-third or middle-third location of esophagus, radionuclide bone scans, and computed tomography (CT) scans from the upper cervical area to the upper abdomen region. Since 2007, whole-body fluorodeoxyglucose positron emission tomography (PET)/CT for esophageal cancer has been a standard routine preoperative staging survey. The pathological stage for patients with esophageal and esophagogastric junction cancer was decided based on the 8th edition of the American Joint Committee on Cancer staging manual. All patients with pT3/T4-stage or positive lymph node metastases after operation in our institution would receive adjuvant therapy accordingly.

### 2.2. Follow-Up Protocol

During the first 2 years in this study, all patients were under active surveillance routinely every 3 months, then and every 6–12 months thereafter. Examinations in the follow-up program included a serum tumor marker test, plain radiography of chest, and a chest CT scan which were routinely arranged. Several exams were conducted if clinically indicated such as esophagogastroduodenoscopy, an abdominal ultrasound, brain magnetic resonance imaging (MRI), radionuclide bone scans, and a PET/CT scan. These examination results and treatment plans have been discussed at regular multidisciplinary team meetings since 2010.

### 2.3. Multiple Primary Cancers

Warren and Gates [12] described that the definition of multiple primary cancers (MPCs) is focused on the following criteria: (1) tumors clearly malignant on histological examination; (2) isolation of tumors by normal mucosa; and (3) the possibility that a second tumor may originate from primary tumor (metastasis) must be excluded. All patients with MPCs had received tissue proof and met the criteria in our study. Regarding the timings of esophageal lesions, MPCs are classified as antecedent, synchronous, and subsequent. MPCs are “synchronous” if diagnosed within 6 months of a diagnosis of esophageal cancer. Otherwise, MPCs more than 6 months before esophageal cancer are defined as “antecedent”, and “subsequent” if diagnosed more than 6 months later.

### 2.4. Statistical Analyses

A chi-squared test was conducted to compare categorical variables and an independent *t*-test and ANOVA test were carried out for the comparison of continuous variables. In the survival analysis, OS was defined as the period between curative esophagectomy and death or the last follow-up. Survival curves were estimated by using the Kaplan–Meier method and the differences between strata were compared by means of the log-rank test. Cox’s proportional-hazards model for the prognostic factors was used. All calculations were performed using SPSS 25.0 software (IBM Corporation, Armonk, NY, USA), and a two-sided *p*-value < 0.05 was considered significant. The picture design was based on R version 4.1.1 with the aid of the Survival, ggplot2, and survminer packages. (The R Foundation for Statistical Computing, Vanderbilt University, Nashville, TN, USA).

## 3. Results

### 3.1. Patient Characteristics

Of 397 patients who underwent curative esophagectomy and were included in our cohort, 101 (25.4%) were associated with multiple primary malignancies. Among patients with MPCs, 29 (7.3%) encountered antecedent cancers before esophageal SCC, and 23 (5.8%) were associated with synchronous cancers. After curative esophagectomy, 49 patients (12.3%) developed subsequent second primary cancers within 5 years. Patients who had no synchronous or metachronous cancer were classified as the “control” group (Figure 1). Clinical features and treatment modalities are shown in Table 1. No substantial differences were found in terms of age, sex, smoking (>1 year), alcohol consumption (>1 year), and clinical stage. The antecedent group all had comorbidity of solid tumors or malignancies (*p* < 0.001), and after neoadjuvant therapy more were observed (*p* = 0.048). Furthermore, there were significant differences in the surgical method for reconstruction and in the reconstruction route (*p* < 0.001 and *p* = 0.006, respectively), mostly by the surgeon’s judgment or preference.

### 3.2. Long-Term Survival and Cox Regression Model

The 5-year OS rate was 44.6% in our cohort, 42.3% in the control group, 36.4% in the antecedent group, 41.7% in the synchronous group, and 65.2% in the subsequent group, respectively. The median observation time was 44.2 months (interquartile range (IQR): 17.7–81.6 months). There were no statistically significant differences in OS between the control, antecedent, and synchronous groups, and patients with subsequent cancers had significantly longer OS (*p* = 0.005) than did those without any MPCs (control group) (Figure 2A). There were no significant differences found (*p* = 0.599) between patients encountering antecedent/synchronous tumors initially and those with no MPCs (Figure 2B).

In Table 2, multivariate Cox regression analysis showed that sex (hazard ratio 1.62, 95% CI: 1.05–2.51, *p* = 0.031) and clinical stage III/IV (hazard ratio 2.43, 95% CI: 1.90–3.11, *p* < 0.001) were independent prognostic factors for OS. The effect on OS was not statistically identified for antecedent/synchronous cancers, older age (≥65), comorbidity, neoadjuvant therapy before surgery, smoking, or alcohol consumption.

### 3.3. Site of Multiple Primary Cancers and Prognosis

A total of 105 MPCs were observed in 101 patients (Table 3). Double primary cancers were identified in 97 patients, and triple primary cancers in four patients. The most frequently observed site of MPCs was the head and neck (65.7%, 69/105), followed by the liver (7.6%, 8/105) and lung (6.7%, 7/105). Of 101 patients with MPCs, patients associated with head and neck cancer (*n* = 69) did not have a significantly worse OS (*p* = 0.843) than did those without head and neck cancer (*n* = 32) (Figure 3A). Among those patients with antecedent or synchronous cancers, there was no statistical difference (*p* = 0.470) in the survival comparison between patients with head and neck cancer (*n* = 34) and those without (*n* = 18) (Figure 3B).

## 4. Discussion

In this study, we retrospectively reviewed the clinical and prognostic characteristics of patients with MPCs who had received resection of esophageal SCC. For reducing survivorship bias, the presence of metachronous cancers was excluded if 5 years beyond esophageal SCC being diagnosed. Among the cohort in our study, a high incidence rate of MPCs (24.5%) within 5 years was observed. The cause of multiple primary malignant neoplasms is not yet elucidated, which might relate to multifaceted factors and lifestyle choice, including long-term radiation exposure and industrial pollution in the surroundings. The risk of cancer development can correlate with genetic accumulation and/or epigenetic alterations in normal-appearing tissue [13]. It will be increasingly susceptible to MPCs while the mucosal epithelium of the head and neck, lung, or esophagus is exposed to the same environmental carcinogens [14]. Widespread exposures to carcinogens (cigarette smoking and alcohol drinking) are strong risk factors for multifocal areas of precancerous development, including the upper aerodigestive tract (tongue, mouth, pharynx, esophagus, and larynx), and should be taken into account when effects of occupational exposure are addressed [15].

The high frequency of MPCs in patients with esophageal cancer might complicate procedures and initial treatment. Long-term survival and clinical outcomes could be influenced by the effect of MPCs. In Baba et al.’s study [7], patients with early-stage esophageal cancers and synchronous cancer demonstrated independent prognostic factors. There was significantly shorter OS (log-rank *p* = 0.032) in patients with synchronous cancer than in those without MPCs, with the same result when comparing with patients who had experienced metachronous cancer. In contrast, Otowa et al. [6] reported a cohort of 273 patients with esophageal SCC in Japan with no significant differences between those with antecedent/synchronous cancers and those without initial cancers (*p* = 0.826) in survival results. In our present study, survival differences were not seen between antecedent, synchronous multiple cancers, and only esophageal SCC in patients. Even in multivariate analysis, patients with previous or synchronous cancers did not have poorer clinical outcomes. This may have contributed to the surgical curability (R0 resection for esophageal SCC) being similar among patients. Prognostic factors in our cohort revealed that male gender and advanced stage of esophageal SCC suffered worse OS. Based on the International Agency for Research on Cancer [1], approximately 70% of esophageal cancer patients are men, and there is a twofold to threefold difference in mortality rates in comparison with women.

According to a nationwide database study in Taiwan [16], patients with esophageal cancer who received major surgery were expected to be younger or at earlier stages. These patients were expected to live longer and might have had higher chances of developing subsequent malignancies, as independent factors for increasing the cumulative incidence of cancers. In our cohort, 12.3% of patients developed subsequent second primary cancers within 5 years after curative esophagectomy. A better prognosis (*p* = 0.005) was demonstrated in patients with subsequent cancers than in those without any MPCs. In a retrospective study analyzing the risk of MPCs [17], Yoshida et al. suggested that neoadjuvant therapy significantly reduced the incidence of postoperative subsequent cancers (*p* = 0.043) and the trend toward contributing to improved survival (*p* = 0.082). However, in the chemo-radiotherapy arm of supplementary analysis in JCOG0502 [18], a higher incidence of second primary malignancies was observed. For both surgical and chemo-radiotherapy arms for clinical early ESCC, the establishment of optimal surveillance planning is encouraged, especially for those with Lugol-voiding lesions associated with the development of subsequent malignancies, mainly observed in head and neck cancer. Early detection of second malignancies allowed less invasive treatment with a favorable outcome [10].

A strong association was reported between esophageal cancer and head and neck cancer. ESCC patients do not always develop multiple cancers in conjunction with head and neck SCC (even if they consume alcohol and cigarettes). Apart from the effects of long-term exposure to carcinogens alone, Muto et al. demonstrated that the critical determinant is documented between drinking and the genetic polymorphism of alcohol-metabolizing enzymes [19]. Acetaldehyde is considered to have a key role in field cancerization of the squamous epithelium [9]. For cases involving synchronous or antecedent head and neck cancers, it has been considered to be a more formidable challenge. Lee et al. demonstrated a dismal 5-year survival rate (9.2%) in patients who had undergone surgery for esophageal cancer associated with head and neck cancers, in comparison with other MPCs [5]. On the other hand, similar clinical results [20] were reported after esophagectomy in patients with or without head and neck cancer. Among the different sites of MPCs, we also showed no difference after esophagectomy in patients with or without head and neck cancers (*p* = 0.843). Chen et al.’s study [21] suggested that patients with esophageal SCC with synchronous or metachronous head and neck cancer might deserve an acceptable treatment prognosis. Especially for patients with early clinical stage esophageal SCC who had the chance to receive surgery, an appropriate treatment strategy via a multidisciplinary team to improve the outcomes of these diseases was necessary [21].

There are several limitations in our present study. First, this is a single-institution retrospective study with limited cases following the selection criteria. Second, there is a lack of standardized surgical techniques, customized treatment strategies, and even regular multidisciplinary team meetings. Whole-body PET/CT for esophageal cancer was not a standard routine preoperative staging survey until 2007 which may have led to undetected MPCs. Third, the diversity of each multiple primary malignancy could not be documented via detailed description. Finally, we selected only patients with curative esophagectomy and MPCs occurring within 5 years, which may cause selection bias or lack of long-term cumulative incidence analysis.

## 5. Conclusions

Patients with antecedent/synchronous MPCs have acceptable therapeutic outcomes if curative esophagectomy can be achieved. As the most frequently observed site with regard to esophageal SCC, head and neck cancer showed no inferior outcome in comparison to others. Subsequent second primary cancers were often observed in esophageal SCC, and optimal surveillance planning for early detection should be established.

## Figures and Tables

**Figure 1 cancers-14-05263-f001:**
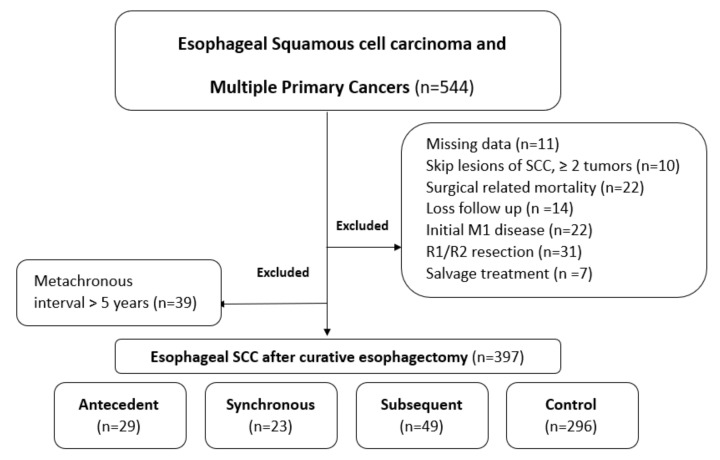
Summary of patient flowchart.

**Figure 2 cancers-14-05263-f002:**
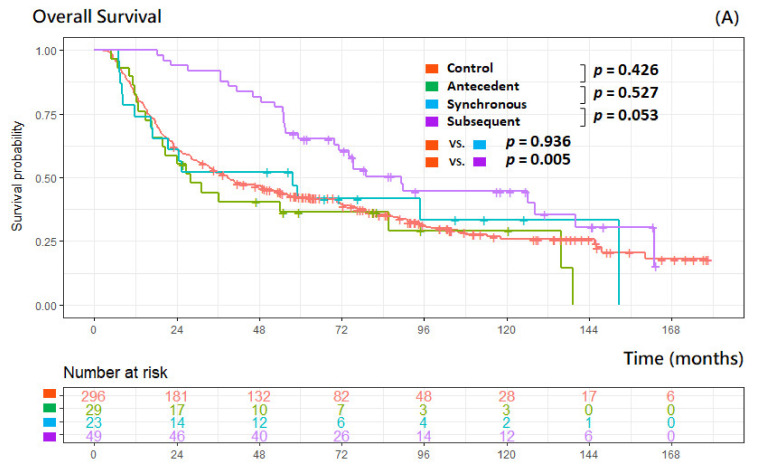
Kaplan–Meier curves according to multiple primary cancer status (**A**), antecedent/synchronous and no MPCs (control) group (**B**).

**Figure 3 cancers-14-05263-f003:**
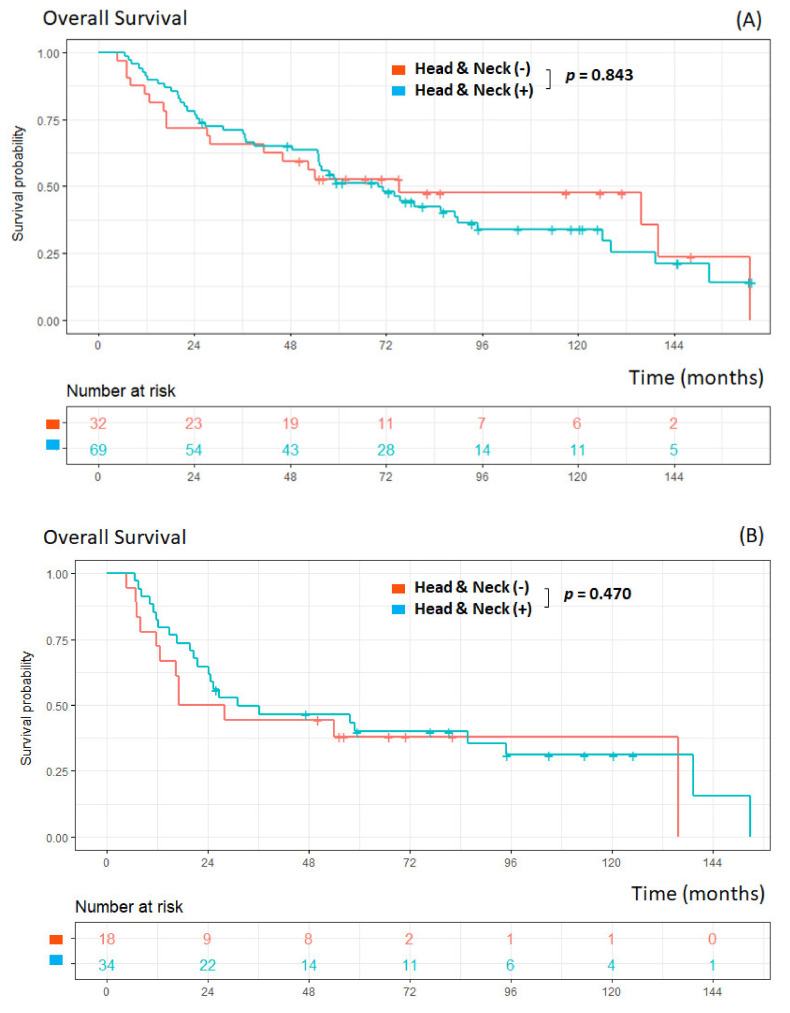
Kaplan–Meier curves with or without head and neck cancer in all MPCs (**A**), in antecedent or synchronous MPCs (**B**).

**Table 1 cancers-14-05263-t001:** Characteristics of multiple primary cancers and control group.

Demographic and Clinical		Metachronous			
Characteristics	Synchronous (*n* = 23)	Antecedent (*n* = 29)	Subsequent (*n* = 49)	Control (*n* = 296)	*p*-Value
Age, median (IQR)	58 (51–62)	57 (52–67.5)	53 (47–62.5)	60 (54–69)	0.102
Sex					0.723
Male	20	27	45	260	
Female	3	2	4	36	
Clinical stage					0.848
I	3	5	10	42	
II	9	12	19	108	
III	9	10	20	115	
IV	2	2	0	31	
Treatment modality					0.048
Neoadjuvant + OP	9	16	15	90	
Upfront surgery	14	13	34	206	
Surgical procedure					<0.001
McKeown	20	24	49	290	
Ivor Lewis	0	4	0	3	
TPLE	2	0	0	1	
RTG	1	1	0	2	
Reconstruction route					0.006
Retrosternal	18	16	42	241	
Post-mediastinum	5	13	7	55	
Comorbidity					<0.001
Absent	7	0	31	191	
Present	16	29	18	105	
ECOG performance status					0.514
0	20	26	45	250	
1	3	3	4	46	
Smoking history					0.389
Yes	16	24	40	216	
No	7	5	9	80	
Alcohol consumption					0.702
Yes	16	22	38	208	
No	7	7	11	88	

IQR: interquartile range, TPLE: total pharyngo-laryngo-esophagectomy, RTG: radical total gastrectomy, ECOG: Eastern Cooperative Oncology Group.

**Table 2 cancers-14-05263-t002:** Predictors of overall survival in Cox regression hazards model.

Variable	Univariate Analysis		Multivariate Analysis	
	HR (95% CI)	*p*-Value	HR (95% CI)	*p*-Value
Age				
≥65	1.13 (0.88~1.46)	0.339		
<65				
Sex				
Male	1.61 (1.04~2.49)	0.032	1.62 (1.05~2.51)	0.031
Female				
Comorbidity				
Present	1.01 (0.79~1.28)	0.948		
Absent				
Neoadjuvant therapy				
Yes	1.27 (0.97~1.62)	0.082		
No				
Smoking				
Yes	1.14 (0.86~1.50)	0.370		
No				
Alcohol				
Yes	0.87 (0.67~1.13)	0.284		
No				
Clinical stage				
III/IV	2.43 (1.90~3.11)	<0.001	2.43 (1.90~3.11)	<0.001
I/II				
Antecedent/Synchronous				
Present	0.83 (0.58~1.18)	0.292		
Absent				

Forward selection in the multivariable model.

**Table 3 cancers-14-05263-t003:** Distribution of multiple primary cancers.

Organ	Antecedent	Synchronous	Subsequent	Total
Head and neck	21 (20.0%)	13 (12.4%)	35 (33.3%)	69 (65.7%)
Liver	2 (1.9%)	1 (1.0%)	5 (4.7%)	8 (7.6%)
Lung	0	4 (3.8%)	3 (2.9%)	7 (6.7%)
Colon/rectum	0	1 (1.0%)	5 (4.7%)	6 (5.7%)
Urological	2 (1.9%)	1 (1.0%)	1 (1.0%)	4 (3.8%)
Gastric	1 (1.0%)	2 (1.9%)	0	3 (2.9%)
Prostate	2 (1.9%)	0	1 (1.0%)	3 (2.9%)
Thyroid	1 (1.0%)	1 (1.0%)	0	2 (1.9%)
Lymphoma	0	0	1 (1.0%)	1 (1.0%)
Skin	0	0	1 (1.0%)	1 (1.0%)
Breast	0	0	1 (1.0%)	1 (1.0%)

One hundred and five sites were identified in 101 patients.

## Data Availability

The datasets during and/or analyzed during the current study are available from the corresponding author upon reasonable request and IRB approval.

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
