# Peer review of "Overall Survival for Esophageal Squamous Cell Carcinoma with Multiple Primary Cancers after Curative Esophagectomy—A Retrospective Single-Institution Study"

_cancers, 2022, doi:10.3390/cancers14215263_

Round 1

Reviewer 1 Report

Comments to the author

I am grateful for the opportunity to review this interesting manuscript entitled: "Overall survival for esophageal squamous cell carcinoma with multiple primary cancers after curative esophagectomy-A retrospective single institution study". This report is interesting because it focused on the significance of curative esophagectomy for patients with MPCs. However, your manuscript has several major problems as following.

1. You mentioned that surgery should be selected as first-line treatment even with MPCs. I think you cannot suggest it because the comparison among surgery and other treatments was not performed in this study.

2. In your cohort, patients with MPCs from different organs were mixed and the influence of MPCs on the outcomes was confused.

3. I cannot understand why did divide MCPs into three categories, because there was not significant difference among these categories for decision making of surveillance planning. The number of patients in each category was too small according to divide into three categories.

Minor comments

Figure 2A

You should explain the reason why subsequent group was better prognosis than others.

Table 3

You had better also show percentage of each number.

Author Response

Dear Editor,

We are grateful for the constructive comments that originated for the review process. Thanks for the chance for revision. All changes in the manuscript have been marked with yellow color.

Reviewer 1

Comments and Suggestions for Authors

Comments to the author

I am grateful for the opportunity to review this interesting manuscript entitled: "Overall survival for esophageal squamous cell carcinoma with multiple primary cancers after curative esophagectomy-A retrospective single institution study". This report is interesting because it focused on the significance of curative esophagectomy for patients with MPCs. However, your manuscript has several major problems as following.

  1. You mentioned that surgery should be selected as first-line treatment even with MPCs. I think you cannot suggest it because the comparison among surgery and other treatments was not performed in this study.

Response: Thanks for reviewer’s kind comment and advice. We revised the text in our abstract: surgery should be selected as potential therapeutic approach if indicated, even with antecedent/synchronous MPCs on Page 1 Line 19.

  1. In your cohort, patients with MPCs from different organs were mixed and the influence of MPCs on the outcomes was confused.

Response: We sincerely appreciate the kind advice. In our section of Limitation, the diversity of each multiple primary malignancy could not be documented via detailed description. In addition to those patients with heads and neck cancer, the outcomes of patients with MPCs from different organs (limited number) had no significant difference between each other as picture as attached file.

  1. I cannot understand why did divide MPCs into three categories, because there was not significant difference among these categories for decision making of surveillance planning. The number of patients in each category was too small according to divide into three categories.

Response: We sincerely appreciate the kind advice. In patients with esophageal cancer associated with antecedent or synchronous cancers, the treatment strategy might be considered to modify appropriately if necessary. The anatomical proximity or complexity of surgery might influence decision making by inference through the overall rates of curative resection for both esophageal cancer and MPCs. We investigate the prognosis of patient with MPCs in each group after customized treatment.

Minor comments                            

Figure 2A

You should explain the reason why subsequent group was better prognosis than others.

Response: We sincerely appreciate the kind advice. In the discussion section on Page 9 Line 208-214: According to a nationwide database study in Taiwan, patients with esophageal cancer who received major surgery were expected to be younger or at earlier stages. These patients were expected to live longer and might have had higher chances of developing subsequent malignancies, as independent factors for increasing the cumulative incidence of cancers. In our cohort, 12.3% of patients developed subsequent second primary cancers within 5 years after curative esophagectomy. A better prognosis (p = 0.005) was demonstrated in patients with subsequent cancers than in those without any MPCs.

Meanwhile, younger or earlier clinical stages were seemed in our subsequent group even with no statistic difference.

Table 3

You had better also show percentage of each number.   

Response: We thank the reviewer for this pertinent comment. We had added the percentage of each number in Table 3.

Reviewer 2 Report

I read with interest the manuscript

"Overall Survival for Esophageal Squamous Cell Carcinoma 2
with Multiple Primary Cancers after Curative Esophagectomy 3
A Retrospective Single-Institution Study" and am glad for the opportunity to review this study. The study is thoughtfully designed, the manuscript is well written and the conclusions are adequate, howeever I have several concerns:

1. Simple Summary "physicians encounter difficult decision making and even advances in surgical techniques" Should be rephrased, the meaning remains unclear

2. Introduction: Line 52-59: I believe the authors refer to patients with chronic exposure to carcinogens (mainly Alcohol+tabacco), who are at risk of developing MPCs. However the sections is not clear and has to rephrased

3. Material and methods: line 77: Patients lost to Follow-up?

4. Results: How do you explain the very high amount of retrosternal reconstruction which does not seem state of the art in a european view?

5. Results: Please specify Comorbidity, which could be anything from high blood pressure to severe cardiac or pulmonal conditions... ASA score available?

6. Results: Superb survival charts!

7. Discussion: "The epigenetic field cancerization maybe a useful opportunity for cancer
risk evaluation if better understanding and recognition". Please rephrase or leave out!

8. Discussion: "When initial cancer was encountered, there also disclosed no statistical differences in..." Please rephrase... Also: Not relevant refers to perioperative complication rate which was not studied here

9. Study limitations and conclusion: adequate; consider rephrasing "Patients with antecedent/synchronous MPCs deserve curative esophagectomy", which seems a bit dramatic to me

Author Response

Dear Editor,

We are grateful for the constructive comments that originated for the review process. Thanks for the chance for revision. All changes in the manuscript have been marked with yellow color.

Reviewer 2:

Comments and Suggestions for Authors

I read with interest the manuscript

"Overall Survival for Esophageal Squamous Cell Carcinoma with Multiple Primary Cancers after Curative Esophagectomy A Retrospective Single-Institution Study" and am glad for the opportunity to review this study. The study is thoughtfully designed, the manuscript is well written and the conclusions are adequate, however I have several concerns:

  1. Simple Summary "physicians encounter difficult decision making and even advances in surgical techniques" Should be rephrased, the meaning remains unclear

Response: We sincerely appreciate the kind advice. We revised the text of Simple Summary “physicians encounter difficult decision making for appropriate treatment” on Page 1 Line 14.

  1. Introduction: Line 52-59: I believe the authors refer to patients with chronic exposure to carcinogens (mainly Alcohol+tabacco), who are at risk of developing MPCs. However, the sections are not clear and has to rephrased

Response: We sincerely appreciate the kind advice. We rephrased the paragraph “The association of high risk for developing MPCs has been explained by the concept of “field cancerization,” which concerns repeated exposure of the epithelium of the upper aerodigestive tract to carcinogens” on Page 2 Line 52-53.

  1. Material and methods: line 77: Patients lost to Follow-up?

Response: We sincerely appreciate for pointing out the mistake and correct it on Page 2 Line 71.

  1. Results: How do you explain the very high amount of retrosternal reconstruction which does not seem state of the art in a european view?

Response: We sincerely appreciate the kind comment. The route for reconstruction is often debated for requiring esophagectomy for esophageal cancer. As neoadjuvant chemoradiation is currently considered as standard treatment for locally advanced esophageal cancer, avoiding the tumor bed within the radiotherapy field during reconstruction may be crucial for reducing posttreatment complications of the gastric conduit (Ann Thorac Surg 2009;87:400 – 4).In contrast, the use of the posterior mediastinal route results in fewer cardiopulmonary complications and anastomotic leaks, which may be related to its lower rates of hospital mortality (J Chin Med Assoc. 2011 Nov;74(11):505-10.). It still remains no consensus as to the optimal route of reconstruction after esophagectomy.

  1. Results: Please specify Comorbidity, which could be anything from high blood pressure to severe cardiac or pulmonal conditions... ASA score available?

Response: We sincerely appreciate the kind advice. The ECOG performance status was attached in Table 1 which has shown to correlate with response to treatment, quality of life and survival. It plays a key role in treatment decisions and an independent prognostic indicator for patients with advanced malignancy. (Case Rep Oncol. 2019 Sep 25;12(3):728-736.).

  1. Results: Superb survival charts!

Response: Many thanks!

  1. Discussion: "The epigenetic field cancerization maybe a useful opportunity for cancer risk evaluation if better understanding and recognition". Please rephrase or leave out!

Response: We sincerely appreciate the kind advice. The paragraph “The epigenetic field cancerization maybe a useful opportunity for cancer risk evaluation if better understanding and recognition.” was left out.

  1. Discussion: "When initial cancer was encountered, there also disclosed no statistical differences in..." Please rephrase... Also: Not relevant refers to perioperative complication rate which was not studied here

Response: We sincerely appreciate the kind advice. The paragraph “When initial cancer was encountered, there also disclosed no statistical differences in post-operation complications, anastomotic leakage rate or in-hospital death comparing between the two groups” was deleted and left out.

  1. Study limitations and conclusion: adequate; consider rephrasing "Patients with antecedent/synchronous MPCs deserve curative esophagectomy", which seems a bit dramatic to me

Response: We sincerely appreciate the kind advice and comment. We rephrased the paragraph “Patients with antecedent/synchronous MPCs have acceptable therapeutic outcomes if curative esophagectomy could be achieved.” on Page 10 Line 254.